# Experiences of Nurse Managers during the COVID-19 Outbreak in a Selected District Hospital in Limpopo Province, South Africa

**DOI:** 10.3390/healthcare10010076

**Published:** 2021-12-31

**Authors:** Idah Moyo, Siyabulela Eric Mgolozeli, Patrone Rebecca Risenga, Sheilla Hlamalani Mboweni, Livhuwani Tshivhase, Tshimangadzo Selina Mudau, Nthomeni Dorah Ndou, Azwihangwisi Helen Mavhandu-Mudzusi

**Affiliations:** 1Department of Health Studies, School of Social Sciences, College of Human Sciences, University of South Africa, Preller Street, Muckleneuk Ridge, Pretoria 0002, South Africa; mgolose@unisa.ac.za (S.E.M.); risenpr@unisa.ac.za (P.R.R.); mbowesh@unisa.ac.za (S.H.M.); mmudza@unisa.ac.za (A.H.M.-M.); 2HIV Services, Population Services International Zimbabwe, Emerald Office Park, 30 The Chase, Harare, Zimbabwe; 3Department of Nursing Science, School of Health Care Sciences, Sefako Makgatho Health Sciences University, Molotlegi Street, Ga-Rankuwa, Pretoria 0204, South Africa; livhuwani.tshivhase@smu.ac.za (L.T.); selina.mudau@smu.ac.za (T.S.M.); 4Department of Advanced Nursing Science, Faculty of Health Sciences, University of Venda, University Road, Thohoyandou 0950, South Africa; Nthomeni.Ndou@univen.ac.za; 5Office of Graduate Studies and Research, College of Human Sciences, University of South Africa, Preller Street, Muckleneuk Ridge, Pretoria 0002, South Africa

**Keywords:** COVID-19, nurse managers, experiences, challenges

## Abstract

The South African health care system was hard hit by the second wave of Coronavirus disease (COVID-19), which affected nurse managers as healthcare facilities became overwhelmed due to an increased workload emanating from the overflow of admissions. Therefore, this study sought to explore and describe the nurse managers’ experiences during COVID-19 in order to identify gaps and lessons learnt. A descriptive phenomenological research approach was used to explore the experiences of ten nurse managers who were purposively selected from different units of a selected district hospital. Data was collected through telephonic unstructured individual interviews and analysed using Colaizzi’s seven steps method. The study revealed that nurse managers experienced human resource related challenges during COVID-19, worsened by the fact that vacant posts were frozen. It also emerged that there was a shortage of material resources that affected patient care. Nurse managers also indicated that COVID-19 brought a lot of administrative duties plus an additional duty of patient care. Also, nurse managers who had previously contracted COVID-19 experienced stigma and discrimination. The government needs to address resource related challenges in rural public hospitals and provide continuous support to nurse managers, particularly during a pandemic like COVID-19.

## 1. Introduction

The emergence of severe acute respiratory syndrome coronavirus 2 (SARS-CoV-2), known as COVID-19 had a devastating effect on a healthcare system that was already overwhelmed by the burden of disease and shortage of resources [1]. According to the World Health Organisation (WHO) COVID-19 dashboard, as of the 21st of December 2021, there were 274,628,461 million confirmed cases and 5,358,978 million deaths were reported globally [2]. Of these, 6,830,390 million cases and 154,538 deaths were from Africa [2]. At the end of May 2021, after (WHO) declared the COVID-19 outbreak as a public health emergency of international concern, South Africa was one of the countries that was highly affected by COVID-19 in the African region [2,3]. According to Stats South Africa [4], South Africa reported the first case of COVID-19 on 5 March 2020 but has now recorded 3.3 million cases with 90,488 deaths and accounts for 49% of the confirmed cases and 59% of the deaths in Africa. Both rural and urban provinces were affected, with urban provinces highly impacted upon. Since then, nurse managers have been persistently serving as the frontline of defence and the backbone of the healthcare system, with fear and uncertainty of the emergent disease as little was known about it [2].

Nurses and nurse managers are responsible for taking care of COVID-19 patients globally. At the same time, they must manage healthcare facilities, enhance productivity, efficiency, sustainability, and minimize risks [5,6]. The psycho-social impact of COVID-19 cannot be under-estimated and nurse managers might be under pressure and stressed making them vulnerable to mental health problems [7,8,9,10]. The implementation of stringent lock down regulations at various levels, affected patients, nurses and nurse managers worldwide [11]. The aim of lock down regulations in South Africa was to prevent or reduce risks, allow the government and the Department of Health (DoH) to develop emergency preparedness, organise resources and a response plan in order to mitigate the severity of the disaster effectively and efficiently [12]. South Africa was hit hard by the second wave of COVID-19 in December 2020, despite having a response plan. The number of patients admitted in hospitals overwhelmed healthcare facilities. Unlike during the first wave that affected the older generation, this time it was the highly mobile age group of teenagers (15–19 years) that drove the second wave. This period was characterized by partying, drinking of alcohol with no adherence to preventive measures [13].

The second wave hit all the South African provinces, with the resource limited provinces like the Limpopo being the most severely affected. The increased infection rate during this period resulted in a high mortality rate and many health professionals succumbed to the virus [4]. As reported in the media, both locally and internationally, the death of healthcare workers has dire consequences to the delivery of healthcare services to each nation’s population. It is well known that nurses are the backbone of the healthcare system, and the nursing workforce is led by nurse managers whose responsibility is to ensure the delivery of nursing care and perform other managerial tasks for the smooth running of the healthcare system [2,5]. The COVID-19 outbreak came as a shock to all sectors throughout the world and no one was prepared for such a deadly virus. Since it is still a novel disease, many studies are emerging on its behaviour, prevalence, and vaccination, and some have been conducted amongst healthcare workers. However, there are limited studies that have been conducted on the experiences of nurse managers during COVID-19. So, this study aimed to fill this gap by exploring the experiences of nurse managers during COVID-19 in a selected district hospital from a resource limited setting in the Limpopo Province, South Africa. Studies on experiences of nurse managers during the era of COVID-19 are imperative in order to develop strategies that can be used to support them to ensure the smooth running of the hospitals and rendering of quality patient care.

## 2. Materials and Methods

### 2.1. Study Design

This study used descriptive phenomenological research design to explore the experiences of nurse managers during the outbreak of COVID-19. Descriptive phenomenology is a type of phenomenology developed by Husserl, whose philosophy emphasised descriptions of human experience [14,15]. Essentially, phenomenology’s purpose is to illuminate the essence of a person’s experience in relation to a specific phenomenon under study [16,17]. The focus is on providing rich textured description of the individual experiences as described by those who experience it. The role of the researcher is to describe what people experience and how they experience it [18]. In this particular instance, the researchers were interested to understand the meaning of experiences of managers as they executed their management roles during the COVID-19 crisis. In phenomenological research, the aim is to encounter the phenomenon via the individual’s description [19]. A key feature of phenomenological research is ‘bracketing’; the putting aside of one’s own beliefs and knowledge about the phenomenon under study, to avoid personal biases and prejudices influencing data collection and analysis [19,20]. Therefore, in this study the researcher made all efforts of identifying and holding in abeyance preconceived beliefs and opinions about the experiences of the managers at the institution of study.

### 2.2. Settings and Participants

The study setting was a district hospital, with each ward having a bed capacity of about 50 patients and each nurse manger had an average staff establishment of ten to fifteen nurses. Having secured ethical approval for the study, ten participants who met the inclusion criteria were recruited using purposive sampling augmented by snowballing [14]. This facilitated selection of participants by virtue of their capacity to provide richly textured information, relevant to the phenomenon under investigation [21,22]. According to Creswell [23], it is essential that all participants have [similar lived] experience of the phenomenon under study. The inclusion criteria for participating in the study was as follows: operational nurse managers working in Vhembe district who had an experience of having worked during the COVID-19 pandemic and were aged 18 years or older. The hospital manager was contacted to request contact details of operational nurse managers who were running the wards or units in the hospital. They were then contacted by the researcher to inform them about the study and to seek their consent to participate in the study. Those who indicated their interest were then sent emails about the interview information, which included the questions that were going to be asked, time and date of the interview sessions. Thereafter, ten interviews were conducted with a sample of ten nurse managers, which is an acceptable sample size for a phenomenological study. The sample size was determined by saturation, referring to the point at which the data collection process fails to yield new information relevant to the study [24,25,26].

### 2.3. Data Collection

Due to COVID-19 restrictive measures, data collection was done through in depth individual telephonic interviews that were unstructured, from March to May 2021. The researchers used an unstructured interview where a central guiding question was asked to all the study participants. Prompts were determined by what the clients would have said following up to highlight areas that were not clear. Considering that COVID-19 is a novel disease, it was important to explore using an unstructured interview in order to gain insights into the lived experiences of nurse managers from their viewpoints. With the participants’ permission, all interviews were audio recorded. [17]. Every participant was asked a grand tour question: *What have been your experiences in executing your duties during this COVID-19 pandemic*? This was followed by prompts and probing questions urging the participants to elaborate or further expound on something they said. The interviews were conducted in English and lasted between 30 to 60 min, thus allowing each participant enough time to share their experiences of being a nurse manager during the outbreak of COVID-19. The telephonic interviews were recorded in a computer and were transcribed verbatim within 48 h to ensure that the researchers commence with preliminary analysis which assisted in preparing for subsequent interviews and to determine data saturation.

### 2.4. Data Analysis

Colaizzi’s method of data analysis was used to analyse the interview data [27]. This approach follows seven steps approach as outlined below:

In step one, each transcript was read several times to gain a sense of the whole content. During this stage, any thoughts, feelings, and ideas that arose from the researchers due to their experiences in interacting with nurse managers were bracketed. This facilitated the exploration of the phenomenon as experienced by study participants themselves [28,29,30]. On step two, significant statements and phrases pertaining to resource challenges experienced by managers were extracted from each transcript. These extractions were given to two researchers for checking to obtain clarity of thoughts. Suggestions were incorporated. They then compared their analysis and reached a consensus [28,29,30]. During step three, meanings were formulated from the significant statements and discussed by the two researchers. Similarly, both researchers compared the formulated meanings with the original meanings maintaining the consistency of descriptions. Minimal differences were found between the two researchers. Thereafter, an experienced researcher (A.H.M) who established that the process had been done well and that the meanings were consistent checked the whole statements and their meanings [28,31]. The researchers proceeded to step four where the formulated meanings were arranged into clusters of themes, that were shrunk into emergent themes [29]. Following this, both researchers compared their clusters of themes, and a researcher experienced in qualitative research checked the accuracy of these themes. In step five, all emergent themes were defined in an exhaustive description of the phenomena under study. This was achieved by combining all the theme clusters, emergent themes, and formulated meanings into a description to create an overall structure. This was followed by seeking the expertise of the experienced (A.H.M) researcher who reviewed the findings in terms of richness and completeness to provide sufficient description and to confirm that the exhaustive description reflected the resource challenges experienced by nurse managers during the COVID-19 crisis [30]. Next was the sixth, where the fundamental structure of the phenomenon was constructed into an exhaustive description. This process involved reviewing the exhaustive description to identify key elements that were then transposed into a definition of the phenomenon under study, that is, that it reflected the participants’ descriptions of their experiences [32]. This culminated in the final step, that is, step seven, where the exhaustive description was validated with each of the participants by using “member checking” technique. This was achieved by providing the participants with research findings and discussing the results with them. This was done through a follow up telephone interview and participants confirmed that the findings reflected their feelings and experiences [14].

### 2.5. Ethical Consideration

Ethical approval for this study was granted by the College of Human Sciences Research Ethics Higher Committee at the University of South Africa (ref number: 90187598_CREC_CHS_2021). Permission to conduct the research study was also obtained from the Limpopo Provincial Department of Health, Vhembe district and the Chief Executive Officer of the district hospital. The voluntary and confidential nature of the study was explained to participants before each interview and consent was obtained for recording of the interview process. To enhance confidentiality, pseudonyms, and age range instead of exact age and years in nursing or management were used in the study. In addition, the hospital name was not mentioned.

### 2.6. Trustworthiness

To ensure trustworthiness and rigour, measures were taken [14]. Trustworthiness refers to the degree of confidence in data, interpretation, and the methodology used to confirm the quality of a study [33]. A reflective diary was kept ensuring neutrality and objectivity. Prolonged engagement, data triangulation, peer evaluation and a co-coder were used to enhance credibility. Records were kept conscientiously locked in a cupboard. This way, an audit trail was left to be followed by any researcher. As part of the evaluation, peer debriefing was done, and irregularities identified were addressed. By coding and recording many times, comparing the themes and categories with a co-coder, researchers ensured dependability. To ensure the analysis accurately reflected the participants’ lived experiences and to enhance authenticity, verbatim extracts from the interviews were utilised [34]. During the generation of themes, the researchers had a peer review process with a senior researcher (A.H.M) who reviewed and challenged the analysis to assure the rigor and credibility of the data analysis [35].

## 3. Results

Ten hospital managers between the ages of 40–65 years formed the participants of this study. The participants comprised nine females and one male nurse manager. All the study participants had more than six years of management experience, had been nurses for 10–40 years, received training on COVID-19 management and had been professionally trained as nurses. Table 1 displays the characteristics of study participants.

Data analysis resulted in four themes with related sub-themes, as presented in Table 2.

The following themes describe the challenges experienced by nurse managers during the COVID-19 outbreak.

### 3.1. Theme 1: Human Resource Related Challenges

Nurse managers indicated that there were human resource related challenges during COVID-19, which made it difficult to allocate staff, to allow nurses who had tested for COVID-19 to work while waiting for their results. What made human resources a serious problem was also the freezing of posts.

#### 3.1.1. Subtheme 1.1: Allocation of Limited Staff Members

Some nurse managers indicated how difficult it was to allocate the limited staff available. Asking for more staff from other units or wards seemed to be a more viable option. However, after a thorough analysis, it emerged that the situation was the same for all hospital wards and thus there was no other options available. The quotation below is in line with the subtheme.
“*I could not make an alternative allocation because there was a huge shortage of staff everywhere. I could not even ask other wards to use their staff temporarily or have a substitute from other wards*.”(Vele)

Challenges related to limited staff were followed by keeping nurses on duty regardless of having tested for COVID-19 and awaiting their results.

#### 3.1.2. Subtheme 1.2: Nurses Working While Tested for COVID-19 and Waiting for the Results

The other problem experienced related to ensuring that nurses who were sick and tested for COVID-19 remained at work until they received their results. One nurse remained working in the ward regardless of her COVID-19 results which came back positive. Nurse managers said:
“*You know, it was so bad because nurses went for testing, but I couldn’t let them go before knowing their results because we were short staffed it was worse when they had to go quarantine at home. There was just so much shortage in the ward*.”(Vele)
“*I mean, I could not call other managers and ask for staff because they also had the same problem I was facing in my ward. Staff shortage is still a problem in our hospital, our nurses are not enough*.”(Ndanga)
“*I also tested but there was no one to relieve me. I could not go home before receiving my results, I had to stay because there was just no one I could leave to manage the ward. We are running short of staff*.”(Vule)

The human resource problem was aggravated by the freezing of posts in that hospital as alluded to below.

#### 3.1.3. Subtheme 1.3: Freezing of Vacant Posts

Nurse managers pointed out that the freezing of vacant posts due to financial constraints in government institutions created more shortages in different hospital wards. The quotations below are in line with the subtheme.
“*We were told all vacant positions had been frozen, meaning that they could not advertise or employ anyone because the government does not have the money*.”(Vele)
“*There are many positions for all nursing categories that are supposed to be advertised and filled and they have been there for a very long time. That’s the main reason we are experiencing severe staff shortage because they just decided to freeze them*.”(Maemu)
“*Government is not hiring nurses; we were just told that all positions were frozen and yet we are struggling because we are working with very few nurses in our wards*.”(Londi)

Unfortunately, all these problems have implications on the nurse manager’s psychological health and the ward environment as pointed out in the next subtheme.

#### 3.1.4. Subtheme 1.4: Implications of the Shortage of Staff on Nurse Managers and the Ward

Some nurse managers showed that the shortage of human resource affected their health and that of other staff members leading to stress. It was not only the nurses who were short staffed, even general workers were, making the ward environment not conducive to patient care. In addition, there was a shortage of drivers.
“*Personally, I felt there was a lot of stress because there was a huge shortage of staff eeh because we are short staffed, you find that there is only two in the ward instead of having six per shift. This affected patient care*.”(Ndanga)
“*We also experienced shortage of general workers in the hospital there are few cleaners and drivers. In some instances, it was difficult to keep the hospital clean*.”(Ndidzu)

The human resource problems were coupled with a shortage of material resources. The second theme illustrates the challenges experienced by nurse managers in that regard.

### 3.2. Theme 2: Material Resources during COVID-19 Era in the Ward

The shortage of material resources was also an outcry for nurse managers. This included lack of oxygen, CPAP machines, drugs, life support equipment, and shortage of linen, water, and PPE. Subthemes have been presented with quotations supporting them.

#### 3.2.1. Subtheme 2.1: Lack of Oxygen

Nurse managers reflected that there was shortage of oxygen, CPAP machines making patient care more difficult especially in cases where a continuous supply of oxygen was required such as in COVID-19 positive patients who had difficulty in breathing. This made nurse managers to run around to the other wards looking for oxygen cylinders to assist patients. The subtheme was supported by the following quotations.
“*We did not have oxygen. There were no CPAP machines and high flow to provide oxygen to patients. Mind you, patients needed to be on continuous oxygen due to COVID-19 related difficulty in breathing. Honestly, how can you save a patient who needs oxygen if you don’t have good supply of oxygen? It was very difficult*.”(Vule)
“*We experienced a shortage of oxygen in the entire hospital, and it was very difficult for patients who were supposed to be on continuous oxygen*.”(Ndidzu)
“*There was a time where we had to run the wards without oxygen heey it was such a nightmare. We had to use oxygen cylinders to support patients who needed continuous oxygen*.”(Londi)

Lack of oxygen was also accompanied by a shortage of medicines and other life support equipment. The subtheme below highlights those challenges.

#### 3.2.2. Subtheme 2.2: Lack of Drugs or Life Support Equipment

Other challenges experienced by nurse managers during the COVID-19 pandemic included shortage of drugs such as antibiotics even before COVID-19 but by then it was worse. Some of the equipment that was not available included vacolitres and intravenous lines as well as immune boosters for both patients and nurses. The quotations below are in line with the subtheme.
“*There are no intravenous lines, vacolitres, no drugs such as antibiotics. We do not have enough medicines for patients, and this has been a problem for a very long time even before COVID-19. This pandemic has worsened the shortage of stock and equipment in the whole hospital. Sometimes, there are no feeding tubes and patients are dying due to this lack of equipment. You know, some of this equipment is very critical for basic life support and it is very difficult to work without them you just can’t save patients*.”(Londi)
“*There were no immune boosters for nurses and the ones for patients were not enough. Sometimes we run short of antibiotics and when you go order in the hospital pharmacy, they will just tell you that there is no stock*.”(Lufuno)
“*Some patients needed intravenous lines and drugs, but we did not have enough. Yes, we tried to ask from other wards and use those that were in our emergency trolley, but they were just enough*.”(Maemu)

Lack of medicines and life support equipment were followed by a shortage of linen presented in the next subtheme.

#### 3.2.3. Subtheme 2.3: Shortage of Linen

The hospital experienced a shortage of linen during this era, which made the nurse managers to start instructing nurses to reuse the dirty linen by patients with the risk of infection. This made the nurse managers to fail to adhere to infection control principles in the wards by allowing patients to reuse the dirty linen. This subtheme has been supported by the quotations below.
“*We experienced shortage of linen because laundry had no one, so we had to re-use dirty linen and mind you it is very old linen. So, aspects of infection control were not adhered to, patients had to use and reuse the very same linen for some time. I mean wrapping themselves in dirty linen*.”(Londi)
“*We don’t have enough linen in the hospital and the one we keep in the wards is very old. We often rely on laundry because they assist in washing it, but it was very difficult during the lockdown because laundry staff were not there, so there was no one to assist us with clean linen. Patients had to use dirty linen re-using and re-using the same sheets and blankets. It was a very big challenge for infection control*.”(Taki)
“*There is no linen it’s old and not enough for our wards if they can provide us with new linen for our ward so that nurses can nurse patients with new and clean linen*.”(Koni)

The challenges of linen were coupled with a shortage of water as mentioned in the next subtheme.

#### 3.2.4. Subtheme 2.4: Shortage of Water

Nurse managers showed that there was also shortage of water in the wards, which made it difficult for them to bath and feed patients. “There were also times where there was no water in the hospital and by that time, we were expected to bath and feed them. Nurse managers had to bring water from home for drinking purposes. Even this was not enough. The shortage of water also affected ablution facilities which makes hospital environment unbearable for both patients and hospital staff. The quotations below from nurse managers support the subtheme.
“*How do you bath and care for patients without water?*”(Mashudu)
“*You know, because we are a rural hospital, sometimes there is no water because of some municipality problems. You find that there is no water in the hospital, and we experienced this during COVID-19, and it was very challenging because this was a critical time where we needed it most*.”(Ndidzu)
“*We often experienced water cut because the pumping system had failed. We had to bring water from our homes to drink by that time patients also needed water to bath and people in the kitchen also needed it to prepare food for patients. Our ablution blocks also needed water eeh sanitation needs water, so when there is no water in the hospital then it becomes a serious challenge for everyone*.”(Maemu)

Water challenges in hospital wards, combined with lack of PPE as discussed in the subtheme below were another problem.

#### 3.2.5. Subtheme 2.5: Shortage/Lack of PPE

Lack of PPE was one of the problems identified by nurse managers. This included a shortage of gowns and general wards were not given any gowns, but instead maternity wards were a priority in the list. This made nurses in the wards to blame the nurse managers for shortage of PPE. The following quotations support the subtheme.
“*There were shortages of PPE and we were only given one gown for two days which contributed to cross infection. For additional protection, we used surgical marks. In addition, this made us more scared of assisting patients*.”(Ndanga)
“*We had a challenge of PPE stock and there was huge shortage of gowns. Sometimes general wards were not given PPE and it was only distributed to maternity and COVID-19 ward. We had to run around requesting supplies such as gowns to protect ourselves…it was worse when we had death because we could not even try to touch the corpse without PPE*.”(Taki)
“*We did not have PPE. We had to fight with people and when I go to the ward, I was blamed by staff for not giving them PPE*.”(Vule)

The shortage of human resources discussed above led to an increased workload as explained in the next theme.

### 3.3. Theme 3: Increased Workload

Nurse managers stated that COVID-19 had brought a lot of administrative work with the additional duty of patient care. It consists of two subthemes as outlined below.

#### 3.3.1. Subtheme 3.1: More Administrative Work for Nurse Managers

A few nurse managers indicated that the management of COVID-19 had brought about many administrative issues. These included screening of all staff members and visitors, recording of activities done in the ward as well as tracing contacts of all patients that had tested positive for COVID-19. The changes made from how things were run before created difficulty in nurse manager’s ability to cope with the additional administrative work. Quotations supporting the subtheme have been presented as follows:
“*This increased our workload, and we are not coping. You know I was the only one calling staff to ensure that everything was running smooth and that everyone was ok*.”(Koni)
“*There is an increase in our workload due to a lot of administrative work from government and we have to comply to the regulations. We have to do daily screening for all workers and visitors every day or every time one enters the hospital and mind you everything has to be thoroughly documented so that we can make follow-ups if there are positive cases who should be traced as a result of contact. This on its own increases the workload to the already compromised staff I mean we are far from being enough*.”(Mashudu)
“*This has resulted in changes of the way we do things and there is so much work that has been added to us*.”(Lufuno)

Nurse managers were not only having administrative work as an additional workload, but also patient care was another duty added to them. See the following subtheme.

#### 3.3.2. Subtheme 3.2: Patient Care by Nurse Managers

Nurse managers had an additional duty to their work roles due to COVID-19 related activities. Apart from providing patient care, nurse managers were expected to provide COVID-19 related counselling to other staff members. The following quotations were cited by nurse managers.
“*I had to run around looking for staff and equipment to use in the ward just to ensure that that the ward was running. This has increased our workload. You know I’m the manager but at times I had to assist with patient care because there were no nurses. It’s not easy to multitask, the workload is too much. It feels like an added burden that will run forever*.”(Taki)
“*You know I had to counsel nurses and cleaners in the ward because they were not coping, and this increased my workload. I think we were all bombarded with a lot of work*.”(Londi)

Nurse managers were working in the COVID-19 ward providing patient care to those patients who had tested positive for COVID-19. Later the nurses themselves also tested positive for COVID-19, creating stigma and discrimination against those nurse managers. See the theme below.

### 3.4. Theme 4: Experiences of Stigma and Discrimination

Nurse managers who had previously tested positive for COVID-19 experienced varying forms of stigma and discrimination as shown below.

#### 3.4.1. Subtheme 4.1: Stigma and Discrimination from Colleagues

Stigma and discrimination were highlighted as one of the problems experienced by nurse managers working in COVID-19 ward as well as after testing positive for COVID-19. Nurse managers showed that when coming back to work after quarantine no one wanted to sit next to them at work to an extent of chasing them from the management meeting. These experiences made nurse managers to feel traumatised as it was done by their colleagues. Quotations below support the subtheme.
“*At first, when one tested positive for COVID-19, there was stigmatisation. When I visited the department after quarantining at home, I was chased away, and they told me that they can’t be with me. It was very traumatizing, and I felt so stigmatized*.”(Vule)
“*There was huge stigma against people with COVID-19 and it was worse for nurses working in COVID-19 ward. I remember someday during our management meeting; I was chased away by the hospital manager, and I was told to go for testing. People did not want to sit next to me during the meeting because I was overseeing the smooth running of that ward. I was chased away. You couldn’t even cough because when you cough then they think you have COVID-19*.”(Koni)

Nurse managers also experienced stigma from support staff as depicted by the following quote.

“*It was so tough because the support staff were also stigmatizing us*.”(Lufuno)

#### 3.4.2. Subtheme 4.2: Stigma from the Community

Nurse managers indicated that they also experienced stigma from the communities where they live. This made nurse managers to feel as if they were carrying and spreading the COVID-19 as people were afraid coming close to them. The quotations support the subtheme.
“*Everyone was just too scared of nurses. They would just run away or avoid you. It felt like we were the ones who were carrying and spreading the virus. In our communities, no one wanted to even come close to us…once they hear that you are a nurse, or you are working in the hospital then they start calling you names*.”(Lufuno)

The extent of stigma extended to the members of the community calling them names due to the fact that they were working in hospitals.

## 4. Discussion

The findings of this study showed that the nurse managers experienced several challenges as they executed their duties during the COVID-19 pandemic. Data analysis yielded four themes, including human and material resource related challenges, increased workload and instances of stigma and discrimination. The first theme on nurse managers’ experiences is “human resource related challenges.” The shortage of human resources demonstrated in this study made allocation of staff extremely difficult. In an effort to maximize on usage of available staff, nurse managers had to come up with measures to enhance availability of staff to provide patient care in the COVID-19 wards. One such strategy used by nurse managers included keeping nurses on duty after testing for COVID-19 until they had received their COVID-19 test results. This was detrimental to the health of those nurses who were providing care to patients whilst they were sick. The human resource challenge was worsened by the fact that all vacant posts were frozen, hence hospitals could not fill these in order to address the problem. Studies [35,36,37] found that making nursing personnel available and adjusting staffing levels in a pandemic such as currently with COVID-19 is crucial.

According to Isobe [38], the healthcare workforce plays a critical role in ensuring an effective functional health care system. However, a shortage of nurses is a major challenge affecting health care systems around the world. Related to this, Poortaghi, Shahmari, and Ghobadi [39] indicate that in an effort to address challenges associated with shortage of staff during the COVID-19 era, some of the strategies utilised included redeploying nurses from other wards to assist in COVID-19 wards. Studies [40,41,42,43] found that nurse managers during the COVID-19 crisis made up for this shortage of staff by recruiting additional staff and ensuring that the hospital had a ready and available number of reserve nurses. This is contrary to the findings of this study where it was difficult or impossible to get relief nurses from other wards since the shortage of staff was affecting all units. The establishment of the nursing technical support teams was found to be an effective strategy during the COVID-19 crisis, and it was responsible for conducting an analysis on hospital workload and provision of related guidance [40]. The nurse managers who were part of this technical team were then able to adjust work schedules and ensure efficiency. Related to this, was ensuring that the hospital had ready and available reserve nurses. Therefore, it is recommended that policy makers, hospital authorities and nurse managers in Vhembe district adopt these strategies that have been proven to be effective.

According to a study by Lam, Kwong, Hung, Pang, and Chien [43], an increased workload and shortage of staff have been found to be limiting factors in curbing the spread of infectious diseases. In addition, such a situation has a negative effect on the physical and mental health of nurses. The shortage of nurses found in this study resulted in nurse managers being stressed as these posed challenges in the execution of their duties. Positive examples from China have demonstrated that the provision of psychological support to frontline health care workers played a critical role during an emergency resulting in positive mental health outcomes [44,45]. The researchers recommend that the health authorities in Vhembe should set up differentiated psychosocial support services specifically for nurse managers working under a difficult and an uncertain environment brought by the COVID-19 pandemic.

The second theme, “shortage of material resources” shows the inadequacy of material resources during the COVID-19 pandemic as experienced by nurse managers. The shortage of these material resources included lack of oxygen, Continuous Positive Airway Pressure (CPAP) machines, medicines, life support equipment, shortage of linen, water, and personal protective equipment. Studies found further gaps in the ability to provide care for respiratory conditions due to lack of pulse oximeter and oxygen tanks [46]. Whilst oxygen is central in the management of admitted COVID-19 patients in respiratory distress, evidence has demonstrated that the adequacy of oxygen is seemingly poor in some sub-Saharan African countries [47]. As was found in this study, this lack of oxygen is concerning. The Médecins Sans Frontiers [48] indicates that the lack of continuous oxygen was experienced in Brazil, Democratic Republic of the Congo, Lesotho, Yemen, and elsewhere in South Africa. Other crucial resources that were a challenge during the pandemic included: shortage of antibiotics, shortage of linen and water, resulting in reuse of dirty linen. This had an effect on the quality of patient care. The findings of the study revealed that the nurse managers experienced a shortage of water due to recurring interruptions to the main local municipal water supply, something very concerning in the running of hospital where patient care mostly depends on water as a basic human need. This observation on water as a basic human need particularly in a healthcare setting has been noted in one study [49]. However, the water challenge is not unique to this hospital, as most rural-based hospitals are under-resourced compared to the hospitals in the urban areas. The discrepancy between urban and rural hospitals has been found in other settings [50,51]. Moreover, due to centralised decision-making, the hospital could not even have its own borehole, which could serve as a backup water supply when the municipality fails. The authors recommend that provincial government and the national department of health to mitigate take measures to address the problem in a more permanent way. Absence of the intravenous fluids is also very crucial for resuscitation of patients who come in dehydrated due to vomiting related to COVID-19 symptoms. A study [52] supports the findings of this study and alluded to the fact that many health institutions are facing major issues such as the critical shortage of beds, and medical supplies, including personal protective equipment.

The findings of this study also indicate that there was inadequacy of PPE in hospitals in Vhembe district and in some instances, there was re-use of PPE. Similar findings were noted elsewhere where it was established that supply chains for personal protective equipment were disrupted, resulting in the need for its reuse [53,54]. Other studies that have examined PPE access during the COVID-19 pandemic also found insufficient supplies in different health settings [55,56]. In that regard, the World Health Organisation [57], indicates that shortage of material resources such as gloves, medical masks, respirators, goggles, face shields, gowns and aprons are leaving healthcare workers inadequately equipped to care for COVID-19 patients. A study conducted in the Republic of Ireland [58] demonstrated that PPE is an important resource in enhancing infection control and prevention practices in hospitals. The study also indicated that healthcare workers should wear protective clothing when there is a risk of contact with blood, body fluids, secretions, and excretions as in the case of COVID-19 [40]. However, this was not the case in this study because risky wards like the one where COVID-19 patients are cared for, staff members had to nurse patients without the relevant PPE. This put the health of nurses at risk of contracting infections including COVID-19.

The third theme, “increased workload,” showed that nurse managers had increased workloads because of the COVID-19 pandemic. A lot of administrative work with the additional duty of patient care brought this about. These administrative duties included overseeing the screening of all staff members and visitors and contact tracing. Due to a shortage of staff, nurse managers also joined the few nurses in the provision of patient care. Apart from providing patient care, nurse managers had to counsel other staff members on matters related to COVID-19. This affected the routine of how things were run before creating more difficulties in nurse manager’s ability to cope. In China, healthcare workers experienced sudden and dramatic challenges in the workplace due to an increased workload, reassignment, and redeployment to other roles [59,60]. In this regard, several studies illustrated that during COVID-19, the healthcare workers were being faced with an increased workload [61,62,63]. Similar findings were noted in the United Kingdom where there were shortages of nursing staff because of an increased workload and unfilled vacancies. Related to this, a study in Iran showed that the shortage of staff was due to reduced nurses’ working hours to [64] reduce the risk of exposure and facilitate flexible shift and rest for nurses [38,65]. However, in the context of this study, the situation was different because the creation of flexible shifts was not possible. Increased workloads were accompanied by experiences of stigma and discrimination described below.

The fourth and final theme, “stigma and discrimination” reflects the participants’ experiences of stigma and discrimination after contracting COVID-19. The findings of this study showed that some nurse managers who had previously tested positive to COVID-19 experienced varying forms of stigma and discrimination from colleagues, support staff as well as from community members. Some nurse managers on coming back to work after quarantine found that no one wanted to sit next to them, to an extent of chasing them from the management meeting which led to them feeling traumatised.

Evidence has demonstrated instances of stigma and discrimination against health care workers [66,67]. Another study [68] concurs with the findings of this study and alluded to the fact that since the outbreak of the COVID-19 pandemic in India, there emerged a negative perception toward those healthcare providers who had previously been infected with the disease. They were being stereotyped as the active spreaders of coronavirus. With such attitudes and the non-supportive environment during the COVID-19 pandemic, the execution of management roles by the participants became difficult. Studies elsewhere have demonstrated that accurate information dissemination plays a key role in creating an enabling environment to combat COVID-19 [67,69]. The COVID-19 pandemic is linked to a rise in stigma and discrimination, which is likely to have a negative impact on mental health, especially when combined with additional outbreak-related stressors [70]. The findings of this study concur with those of the Nepalese one where their fellow nursing colleagues as well as community members who, when sick, expect the nurses in question to provide good patient care to them were treating nurse managers like outcasts. The World Health Organisation [63] calls for the provision of comprehensive support to the healthcare providers and the community to create an enabling environment and to improve the mental health of healthcare providers during the COVID-19 crisis.

## 5. Limitations

The study was limited in that it is phenomenological based, subjective in nature and not data driven like any other qualitative study. However, subjective as it is, it afforded the researchers an opportunity to gain insights into the lived experiences of nurse managers during COVID-19- a novel disease. It was conducted in a district hospital in Limpopo province and hence the findings cannot be generalised to other settings. Data was not collected through face-to-face interviews due to the COVID-19 restrictive measures and some of the physical reactions expressed by participants might, therefore, may have been missed.

## 6. Conclusions

The study findings highlighted the fact that nurse managers had to deal with staff shortages daily. This situation shows the urgent need for improving staffing at the hospital and supporting available nurses in rural public hospitals to ensure that they continue to provide services. There is also a need for the government to provide continuous support to nurse managers, more specifically during a pandemic like COVID-19. What also emerged from this study was the lack of preparedness by the rural hospitals which are also under-resourced. Nurse managers who are currently overwhelmed need urgent departmental support to cope with their work loads. The study underscores the need to put in place response plans to emergencies and to activate disaster plans.

## Figures and Tables

**Table 1 healthcare-10-00076-t001:** Demographic characteristics of participants.

Name	Gender	Age Range	Number of Years in Nursing	Number of Years as Manager	Training in COVID-19 Management	Professional Nurse Training
Vele *	Female	40–45	10–15	6–10 years	Orientation	Four-year Diploma
Ndonga *	Female	56–60	15–20	6–10	Orientation	Dip. in General Nursing
Ndidzu *	Female	56–60	36–40	11–15	Orientation	Dip. in General Nursing
Maemu *	Female	50–55	20–25	6–10	Orientation	Four-year Diploma
Taki *	Female	46–50	15–20	6–10	Orientation	Dip. in General Nursing
Londi *	Female	60–65	31–35	10–15	Orientation	Four-year Diploma
Vule *	Male	50–55	31–35	11–15	Orientation	Dip. in General Nursing
Mashudu *	Female	40–45	15–20	4–8	Orientation	Four-year Diploma
Lufuno *	Female	50–55	31–35	9–14	Orientation	Four-year Diploma
Koni *	Female	40–45	15–20	6–8	Orientation	Four-year Diploma

* Pseudonyms were used.

**Table 2 healthcare-10-00076-t002:** Summary of themes that emerged from data analysis.

Themes	Subthemes
1. Human resource related challenges	1.1 Allocation of limited staff members
	1.2 Nurses working while tested for COVID-19 and waiting for the results
	1.3 Freezing of vacant posts/no employment
	1.4 Implications for shortage of staff on nurse managers and the ward
2. Material resources during COVID-19 era in the ward	2.1 Lack of oxygen
	2.2 Lack of drugs or life support equipment
	2.3 Shortage of linen
	2.4 Shortage of water
	2.5 Shortage/Lack of PPE
3. Increased workload	3.1 More administrative work for nurse managers
	3.2 Patient care by nurse managers
4. Stigma and discrimination	4.1 Stigma and discrimination from colleagues
	4.2 Stigma from the community

## Data Availability

The data presented in this study are available on request from the corresponding author.

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
