# Peer review of "Experiences of Nurse Managers during the COVID-19 Outbreak in a Selected District Hospital in Limpopo Province, South Africa"

_healthcare, 2021, doi:10.3390/healthcare10010076_

Round 1

Reviewer 1 Report

This study by authors explore and describe the nurse managers’ experiences during COVID-19 in order to identify gaps and lessons learnt.

Introduction: Line 42 SARS stands for Severe Acute Respiratory system not Severe Acute Corona syndrome. COVID-19 should be in capital letters at all places including the title.

Overall introduction is well written, highlighting the importance about the topic, discussing the impact of COVID-19 in healthcare settings and impact of lockdown affecting the services provided in the health systems. Introduction also provide readers about rationale of this study.

Methods: Methods are clearly defined, including study designs, ethical clearance was taken. Phenomenological studies have meticulous approach in their design and require rigrous efforts from authors for which I would like to congratulate the team. The Colaizzi’s method is described in its entirety and explain the readers what each step means in regards to the current study.

Results: Table is helpful in describing the demographics. Themes and subthemes are clearly laid out with verbatim examples to show the personal experiences that some of the participants described.

Discussion and conclusion: A well done job in discussion section highlighting themes and its implications. Authors have highlighted the phenomenology of nurse managers during COVID-19 and thus trying to bring important themes that require advance planning and implementation for future emergency and disaster plans.

Limitations: Should additionally include that this is not data driven study and rather phenomenological based study which can have its own bias as portrayal of scenarios by media which can bring a common theme to the discussion point.

Author Response

  • The word covid-19 was put in capital letters as ‘COVID-19’ on the title and throughout the manuscript
  • On the introduction, on line 42, ‘Severe acute corona syndrome (SARS – Co-V 2), was changes to read ‘severe acute respiratory syndrome coronavirus 2 (SARS-Co-V 2’
  • More information was added to outline the limitations of the study

Reviewer 2 Report

Thank you for an opportunity to read this timely paper on the experiences of nurse managers in South Africa during the COVID-19 pandemic. The paper fills an important gap in the literature, and provides a raw and at times shocking record of the burdens placed on nurse managers.

The quotes have been carefully chosen and are very moving.

The paper would benefit from careful proof editing to correct minor errors such as:

  • inconsistency in the use of "Corona virus disease" vs "Acute corona syndrome" rather than the WHO approved term "severe acute respiratory syndrome coronavirus 2" (SARS-CoV-2) or COVID-19 to refer to the disease
  • "3,530,582 million deaths" rather than 3.5 million deaths; the authors should also include the month and year at which these figures were take and perhaps update prior to resubmission
  • separating the introduction into shorter paragraphs
  • I don't think it's necessary to say participants had to be over the age of 18 as the inclusion criteria already restricted participants to nurse managers who would all be adults anyway
  • I suggest using "shortage of staff" rather than "shortage of manpower"

Can the authors explain why they chose an unstructured interview format? Did you have any type of interview schedule for the follow-up prompts?

Can you give some idea of the size of wards eg roughly how many staff and patients would each manager be responsible for?

Could the authors please explain how and why water supplies were affected during the pandemic?

The authors refer to a shortage of gowns. As COVID is an airborne disease, it would be helpful to know what sorts of masks staff were wearing. Did they have surgical masks, N95 masks or fit-tested N95 masks?

I would suggest placing limitations before conclusions to finish strongly.

Author Response

The whole manuscript was proofread and edited, and grammatical errors were corrected.

The inconsistency of “corona virus disease” and “acute respiratory syndrome coronavirus 2” was corrected. The phrase "severe acute corona syndrome (SARS-CoV-2)” was replaced by “severe acute respiratory syndrome coronavirus 2 (SARS – Co-V 2)”

The COVID-19 figures and cumulative confirmed cases and deaths were aligned to the dates when information was retrieved.

The introduction was separated into three paragraphs

To enhance clarity, the statement ‘had to be over the age of 18’ was removed from the inclusion criteria.

As suggested by the reviewer, the term “shortage of manpower" was replaced by "shortage of manpower

More information was added to explain the rationale of using unstructured interview as shown in the manuscript on the tracked section under the data collection section.

Information was added on the estimated number of staff and patients per ward.

More information was added on the challenge associated with supply. We have re-looked at the transcripts and realised that there was something that had been missed.

Reviewer 3 Report

The COVID-19 pandemic is undoubtedly a challenge for the health sector. The last two years have shown that effective counteracting the spread of SARS-CoV-2 virus and combating its effects is beyond the capabilities of many countries, including wealthier ones and with more efficiently organized health systems. A properly functioning health service is a necessary condition for good economic development and the quality of life of the society. The dramatic improvement in current and strategic governance in the health sector during the COVID-19 pandemic is a key message for those managing and providing health services, including nurses. Therefore, the choice of the research topic is principles and timely. The qualitative analysis of the main problems with the functioning of the health system during the pandemic well reflects the current situation of the health care system in South Africa. The authors' research has shown that healthcare is overburdened with staff and economic problems, and that medical workers require psychological support. There is no reference to bibliographic items No.36, 42 and 68 in the text of the work.

Author Response

The bibliographic items number 36,42 and 68 that had been omitted were inserted in the text of the manuscript accordingly (as shown by the tracked changes).

Reviewer 4 Report

The manuscript describes the challenges faced by nursing managers
during the COVID-19 pandemic in South Africa.
The qualitative approach is rich and brings relevant aspects
faced by these professionals. My only suggestion is to review
the Introduction section, some sentences are repetitive.
I suggest including more details about the local healthcare system.

Author Response

The introduction was reviewed and edited to remove repetition as shown by tracked changes.